# Bioinformatic Analysis of the Leptin–Ob-R Interface: Structural Modeling, Thermodynamic Profiling, and Stability in Diverse Microenvironments

**DOI:** 10.3390/ijms26146955

**Published:** 2025-07-20

**Authors:** Gabriel I. Ortega-López, Francisco Reyes-Espinosa, Víctor Eric López-Y-López, Claudia G. Benítez-Cardoza

**Affiliations:** 1Laboratorio de Investigación Bioquímica y Biofísica Computacional, ENMH, Instituto Politécnico Nacional, Gustavo A. Madero, Ciudad de México 07320, Mexico; gortegal2100@alumno.ipn.mx; 2Departamento de Ingeniería Ambiental, Tecnológico Nacional de México/ITS de Comalcalco, Tabasco 86650, Mexico; francisco.reyes@comalcalco.tecnm.mx; 3Laboratorio de Bioprocesos y Biocatálisis, CIBA—Instituto Politécnico Nacional, Sta. Inés Tecuexcomac-Tepetitla, Tlaxcala 90700, Mexico; vlopezyl@ipn.mx

**Keywords:** leptin, leptin receptor, binding free energy change, thermodynamic stability, tissue microenvironment, immunometabolism

## Abstract

Leptin is an adipocyte-derived hormone that orchestrates different physiological processes, including energy balance, thermogenesis, immune regulation, reproduction, and tissue remodeling. These effects are mediated through interaction with the CRH2 domain of the leptin receptor (Ob-R). While the structural aspects of the interaction between leptin and Ob-R have been first studied in humans and mice, comparative analyses of stability across mammalian species under physiologically relevant microenvironmental conditions remain limited. We performed a bioinformatics-driven structural, stability, and thermodynamic characterization of the leptin–CRH2 complex. This included structural homology modeling using a full-length template, interface mapping, and binding energy estimation. Additionally, we analyzed the effect of pH, ionic strength, and temperature on complex formation to mimic physiological and pathological tissue conditions to enhance clarity in the structural features and stability of the complex. Our results show that the interaction is primarily enthalpy-driven and is sensitive to temperature, ionic strength, and pH changes for all heterodimers analyzed here. The predicted binding free energy (ΔG) ranged from −10.50 to −16.81 kcal/mol across species. The integrated bioinformatic analyses suggest that subtle sequence variations influence the stability and environmental responsiveness of the complex. This study provides a molecular framework for understanding how leptin–Ob-R binding adapts across species and tissue contexts.

## 1. Introduction

Leptin is a pleiotropic hormone secreted primarily by adipocytes, involved in regulating appetite, energy expenditure, immune function, and other physiological processes [1,2]. These effects are mediated through binding to the leptin receptor (Ob-R), particularly the CRH2 domain, which initiates receptor activation and downstream signaling cascades [1,3]. Leptin is expressed in adipose tissue, the placenta, and skeletal muscle, among other tissues, while Ob-R is highly expressed in the hypothalamus, gastrointestinal tract, heart, liver, small intestine, prostate, ovary, lung, kidney, and skeletal muscle [4,5,6].

Structurally, leptin is a 16 kDa helical cytokine composed of four antiparallel α-helices (H1–H4) arranged in an up–up–down–down topology, stabilized by a disulfide bond between Cys96 and Cys146, forming a characteristic pierced lasso motif [5,6]. The leptin receptor (Ob-R) comprises several extracellular domains, including the N-terminal domain, cytokine receptor homology (CRH1 and CRH2) domains, an immunoglobulin-like domain, and two fibronectin type III domains. Among these, the CRH2 domain is primarily responsible for high-affinity leptin binding and is essential for receptor activation [7]. It adopts a β-sandwich fold formed by 16 β-strands organized into a sandwich-like topology, facilitating cytokine–receptor interactions similar to those seen in other class I cytokine receptors [8]. And key residues involved in the interaction have been identified [9,10,11,12,13,14,15,16,17,18,19]. Most structural models have focused on the human or murine system, without addressing potential differences across species that could impact binding affinity or stability. Furthermore, the leptin–Ob-R interaction (similar to any other protein–protein complex) might be subject to modulation by tissue-specific microenvironmental factors such as pH, ionic strength, and temperature [20,21].

In healthy tissues, leptin and Ob-R operate under well-regulated microenvironments, where parameters such as pH, ionic strength, and temperature are maintained within narrow physiological ranges [22,23], although there still might be slight variations among tissues or across species [24,25,26,27]. In addition, several conditions, including inflammation, ischemia, infection, and cancer, are characterized by disruptions of these parameters. For instance, acidic pH is a hallmark of tumors and inflamed tissues [28], while elevated temperature and altered ionic environments occur during fever, immune activation, and metabolic stress [20,29,30,31]. Notably, these parameters can vary concomitantly under pathological conditions. The effect of different physicochemical conditions on the formation of the leptin–CRH2 complex has not been fully explored.

Understanding how evolutionary variations in homologous protein sequences among mammalian species affect this interaction and how it responds to different physicochemical conditions is crucial for two main reasons: (1) animal models are widely used to study leptin signaling and therapeutic development, yet structural differences may limit translational relevance and (2) mapping residue-level contributions to interface stability may inform how mutations or engineered substitutions impact receptor binding under diverse physiological or pathological conditions. Here, we present a comparative structural, stability, and thermodynamic analysis of the leptin–CRH2 complex from five mammalian species, chosen based on the availability of fully validated leptin and Ob-R sequences. We model the 1:1 leptin–CRH2 complex, and characterize it by differences in interface interaction and complex stability. Furthermore, we evaluate the effects of pH, ionic strength, and temperature on the stability of binding in mimicking physiological conditions. This work offers new insights into the conservation and divergence of leptin–receptor interaction and its energetic profile and discusses some possible implications for tissue-specific function and cross-species translation.

## 2. Results and Discussion

### 2.1. Sequence Conservation and Structural Consistency

To investigate evolutionary conservation and suitability for structural modeling, we first compiled revised leptin and CRH2 sequences from five species (human, mouse, rat, pig, and macaque) from UniProt and aligned them using Clustal Omega. We then constructed homology models for the leptin–CRH2 complexes from five species using validated templates.

Sequence alignment revealed high conservation among leptin and CRH2 domains across the five species. Leptin showed 73.3% identity (Figure 1A), including the presence of two highly conserved Cysteines (Cys96 and Cys146), which are necessary for the formation of the pierced lasso topology of leptin [10].

The CRH2 alignment of the five Ob-R homologs shows 155 identical positions (77.5% identity) and 26 similar positions (Figure 1B). The human and pig CRH2 sequences have two more amino acid residues than the rest of the sequences studied here. The CRH2 region of the Ob-R (432 to 631 or 430 to 633, based on the biological species) is highly conserved, as previously reported [32].

It is worth noting that, although a crystallographic structure of the leptin–CRH2 complex is available (PDB ID: 7Z3Q), this structure omits 44 residues in leptin and 6 residues in CRH2, and it includes 2 amino acid substitutions in the receptor domain: Asparagine to Glutamine at position 516 (N516Q) and Cysteine to Serine at position 604 (C604S) [33]. Although these residues are not located so close to the leptin-binding interface, these substitutions may influence local electrostatic and hydrogen bonding networks that contribute to complex stability. Due to these limitations, we opted to generate full-length, sequence-validated models for each species via homology modeling using as a template the coordinates of the Protein–Complex Molecular Dynamics Model (PCMDM) (see Appendix A) which corresponds to the average structure of the most representative cluster obtained from molecular dynamics simulations of the human leptin–CRH2 complex previously reported [34]. This model was used as a structural reference for alignment and stability comparison across all species. To evaluate the similarity between the PCMDM and the 7Z3Q crystallographic structure, we performed pairwise 3D alignments (Figure 2). Using YASARA, we obtained an RMSD of 2.083 Å based on 186 aligned residues with 93.01% sequence identity. Using PDBeFold, the RMSD was 2.304 Å over 183 aligned residues with 85% sequence identity. The difference in aligned residue counts reflects variations in the algorithm’s handling of mismatches and gaps. Importantly, not all residues from the crystal structure were used because some segments were missing or poorly resolved, and some interface regions could not be reliably matched. The observed RMSD differences were influenced by specific substitutions of Asparagine516 for Glutamine and Cysteine604 for Serine and the absence of residues 454-IQSLAE-459 in the Ob-R coordinates, as well as by the absence of residues 25-SHTQSVSSKQKVTGLDFIPGLH-46 and 99-PWASGLETLDSLGGVLEASGYS-120 in the coordinates of leptin in the 7ZQ3 structure. Structural similarity, assessed by RMSD [35], is a widely accepted measure in homology-based protein modeling, as it quantifies atomic superposition fidelity and helps evaluate model quality.

This strategy ensured structural completeness and sequence fidelity, allowing us to explore interspecies variations and binding energetics under diverse physicochemical conditions with greater reliability.

All leptin–CRH2 models built by homology showed acceptable values for construction quality in YASARA (RMSD < 0.110 Å) and PDBefold (Q-score = 0.99) [36]. The backbone alignment of the 3D models of the leptin–CRH2 complex did not show any significant structural displacements compared with the PCMDM template (Figure 3A).

### 2.2. Interface Variation Across Species

Following model generation and refinement through molecular dynamics simulations (MDS), we analyzed the interface residues using PDBePISA (EMBL-EBI, Wellcome Genome Campus, Hinxton, UK) and visualized contact patterns using YASARA (YASARA Biosciences GmbH, Vienna, Austria). Comparisons among species highlighted conserved and variable interaction zones.

Interface analysis revealed conserved involvement of leptin helices H1 and H3 and CRH2 loops (βA-βB, βC-βD, βF-βG, βI-βJ, βK-βL, and βO-βP). For example, most of the species have a Lysine at position 5, whereas the pig has Arginine; at position 78, most species harbor an Asparagine, which is substituted by Histidine in leptin in the rat; in human leptin, the leucine at position 89 is substituted by Valine; and at position 92, most of the species analyzed have a Phenylalanine, which is replaced by Serine in the leptin sequence in the pig.

On the receptor side, the main residues involved in the formation of the interface lie within loops (βA-βB, βC-βD, βF-βG, βI-βJ, βK-βL, and βO-βP) in the hinge region. Variable residues modulate contact patterns; for example, most species analyzed contain Tyrosine in loop βA-βB at position 441(439), which is substituted for Histidine in macaque CRH2. In loop βC-βD, residue 467(469) contains Serine in humans, pigs, and macaques, while this position contains Arginine in mouse and rat CRH2 sequences. In the same loop, position 472(474) is quite variable, containing either Serine (human and pig), Proline (mouse and rat), or Phenylalanine (macaque). These sequence variations are relevant in the formation of the interface, its energetics, and its dependence on different physicochemical conditions.

The interface of the 7Z3Q crystallographic model also shows that alpha-helices H1 and H3 of leptin and the six loops (βA-βB, βC-βD, βF-βG, βI-βJ, βK-βL, and βO-βP) of Ob-R are involved, as in the PCMDM. The interface area of the 7Z3Q model is larger (837.7 Å^2^) than that of the PCMDM (790.4 Å^2^). The difference in area between the models is mainly due to the additional amino acids involved in Ob-R in the 7Z3Q model. For example, Thr443, Arg468, Asp475, Gln501, Pro502, Leu530, Asn566, Asn567, and Leu619 of Ob-R and Asp8, Arg71, and Ile74 of leptin do not participate in the PCMDM but do participate in the 7Z3Q model. On the other hand, Ser469, Pro561, Pro564, and Gly618 of Ob-R and Thr10, His88, Phe92, and Ser93 of leptin do participate in the PCMDM but not in the 7Z3Q model.

In Figure 3B–F, the leptin–CRH2 heterodimer models show the corresponding interfaces. The interface area between the complexes studied here varies by about 95 Å^2^. The rat leptin–CRH2 complex shows the largest (885.8 Å^2^) hidden surface, indicating a higher number of contacts, involving 49 amino acids (24 in leptin and 25 in Ob-R). On the other hand, the interface area of the human model is the smallest (790.4 Å^2^), suggesting a reduced number of interactions (44 amino acids; 23 in leptin and 21 in CRH2), similar to previous reports that indicated an interface area of 750 Å^2^ [17]. The mouse, pig, and macaque interface areas are 862.9, 795.1, and 794.5 Å^2^ and involve 48, 43, and 44 residues, respectively. All interface models show how the hinge region of CRH2 forms a hexadactyl shape formed by a 6-limb clamp-like structure that interacts with alpha-helices -1 and -3 of leptin. In the case of the interface of macaque CRH2, residues located at βI-βJ loop D530(532) to L537(539) might not participate in binding since models showed larger distances between residues at this loop and residues of alpha-helix-1 of leptin. Two research groups reported lists of amino acids involved in the interface of human and mouse leptin–CRH2 complexes [17,34]. Their structural and thermodynamic evaluation of human and mouse complexes involved only five CRH2 loops (without the participation of the βO-βP loop); however, subjecting the PCMDM to Molecular Dynamic Simulations (MDSs) showed the involvement of the residues within the βO-βP loop [34]. The β-strand O of human CRH2, which participates in the interactions at the interface, is shorter than the same strand in other species; consequently, the adjacent flexible loop βO-βP in humans appears longer than in CRH2 in the other species.

The differences in the interface of the leptin–CRH2 complexes (Figure 3C–F) are marked as follows: residues that do not participate in the human model but participate in other complexes are indicated by black arrows (Ile76, Gly438(440), Arg466(468), Pro472(474), and Leu536(538) in the mouse; Ile76, Gly438(440), Arg466(468), Gln499(501), Pro 472(474), Thr441(443), Pro500(502), and Leu536(538) in the rat; Gly440(440), Arg468(468), and Leu538(538) in the pig; and Arg466(468), Phe472(474), and Lys558(560) in the macaque), while positions that have substitutions and also participate in the leptin–CRH2 interface are marked with red arrows (Leu89 in all species, Arg467(469) in the mouse, His78 and Arg467(469) in the rat, Arg5 and Ser92 in the pig, and His439(441) in the macaque). The residues that participate in the interface in the human complex but not in complexes in the other models are enclosed in black circles (Cys471(473) in the mouse, Cys471(473), Asp615(617), and Gly616(618) in the rat, Cys473(473), Ser469(469), Gly618(618), and Pro561(561) in the pig, and Gln75, Asp530(532), and Gly616(618) in the macaque). In previous works [18,19], the models used were built via homology modeling of CHR2 using the structure of Granulocyte Colony-Stimulating Factor (G-CSF) as a template, in combination with literature. Similarly to published reports on the residues in the interface, in our case, the template used for CRH2 corresponds to crystallographic data of the leptin receptor (PDB file 3V6O), which was further refined in complex with leptin using MDS [34].

All the interactions at the interface of leptin and the CRH2 domain are important; however, we focused on a few due to their short distance. First, the shortest distances (1.76 to 1.84 Å) correspond to the hydrogen bond formed between the OD2 atom of Aspartic9 and the HH atom of Tyrosine472 (leptin and CRH2, respectively), which is observed at the interface of leptin and CRH2 in humans, mice, pigs, and macaques, but not in rats. The second relevant hydrogen bond involves the OD1 atom of Asparagine82 and the H atom of Leucine505, which are between 2.19 and 2.48 Å at the leptin and CRH2 interfaces in humans, mice, and macaques. Additionally, we detected another hydrogen bond (2.96 Å) and four salt bridges (2.88 to 3.67 Å) involving OD1 and OD2 atoms of Aspartic85 and NH1 and NH2 of Arginine466 at the interface of the macaque leptin–CRH2 complex.

Appendix A lists the intermolecular interactions at the docking interfaces of the five models analyzed here. At 36.5 °C and 37.5 °C, we found 51, 59, 57, 53, and 49 interactions in the human, mouse, rat, pig, and macaque complex models, respectively. These analyses of the interface were conducted in PRODIGY [37,38].

### 2.3. Analysis of the Stability of Leptin–CRH2 Complexes

We calculated the predicted free binding energy (ΔG_b_) using the FoldX Suite server under physiological conditions (pH 7.0, 298 K, 0.15 M ionic strength).

The changes in the predicted binding free energy (ΔG_b_) values associated with the formation of leptin–CRH2 complex models were estimated at pH 7.0, 298 K, and 0.15 M ionic strength, and they are shown in Table 1. The predicted ΔG_b_ values vary from −16.81 kcal/mol (mouse) to −10.50 kcal/mol (rat). This corresponds to variations of 4.3 orders of magnitude of *K_a_* between these species.

Based on our models, the interface of the rat complex involves more amino acids (49) and more interactions (57), but the change in its free binding energy is weaker than that of the other complexes. The complex with the highest binding energy (−16.81 Kcal/mol) and most docking interactions (59) is the mouse complex, although the SASA (862.9 Å^2^) and the number of amino acids (48) at its interface are similar to those of the rat complex (885.8 Å^2^ and 49). Therefore, a higher number of amino acids at the interface does not guarantee a higher interaction energy or stability, since the distances between residues and the type of interactions must also be considered. For the human leptin–Ob-R complex, the predicted Kd (~13.4 nM) closely aligns with experimentally reported values ranging from 1 to 15.4 nM [39], supporting the validity of our model. In contrast, for the mouse complex, our predicted Kd (~460 nM) is higher than reported experimental estimates (0.1 to 17 nM [16,39]), likely due to dynamic differences in flexible regions and method-inherent approximations. Further analyses must be conducted to explore these differences. As no experimental binding constants have been reported for pig, rat, or macaque leptin–CRH2 complexes, our predictions should be interpreted as comparative indicators of interaction stability rather than definitive binding affinities.

### 2.4. Effects of Physicochemical Conditions on Leptin–CRH2 Complex Stability

We next simulated how changes in microenvironmental conditions (pH, ionic strength, and temperature) affect the thermodynamic stability of leptin–CRH2 complexes across species. These conditions were selected to reflect physiological and pathological tissue variation.

The effects of pH, ionic strength, and temperature change on the predicted ΔG_b_ are shown in Figure 4A,B and Figure 5A, respectively. Although the changes in predicted ΔG_b_ may appear modest, they are not trivial, since a shift of just 1 Kcal/mol in predicted ΔG_b_ at 298 K corresponds to an approximate 5.4-fold change in the equilibrium constant (*K_a_*) [40]. This reflects substantial alterations in the distribution of bound versus unbound species, emphasizing the physiological relevance of these thermodynamic variations. The predicted *K_a_* values of the leptin–CRH2 complexes of the five mammalian species under different physicochemical conditions are shown in Table 2.

#### 2.4.1. Effect of pH on Binding Stability

The predicted ΔG_b_ of the leptin–CRH2 complexes was evaluated across a range of pH values to mimic physiological and pathological conditions in different tissues. As shown in Figure 4A, predicted ΔG_b_ varies based on the isoelectric points (pI) of leptin and CRH2, as well as the protonation state of interface residues.

**Figure 4 ijms-26-06955-f004:**
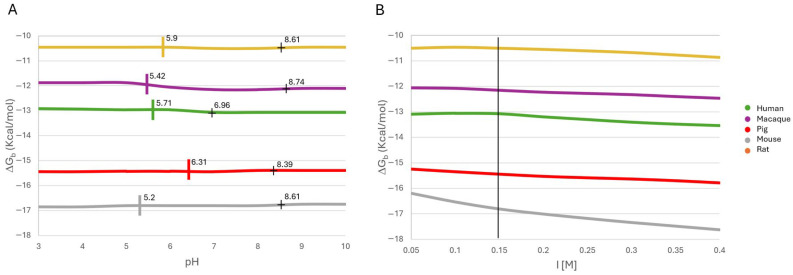
Effect of changes in pH (**A**) and ionic strength, I, (**B**) on the stability of leptin–CRH2 heterodimer models of five mammalian species. The vertical lines in (**A**) indicate the pI values of leptin (wide colored lines) and the CRH2 of Ob-R (black crosses), and the vertical lines in (**B**) indicate the physiological ionic strength.

Complexes from humans and macaques exhibit similar profiles, showing the highest predicted ΔG_b_ values (indicative of reduced complex stability) under acidic conditions. Stability increases, reflected by more negative predicted ΔG_b_ values, at pH levels above the pI of leptin. In contrast, the rat leptin–CRH2 complex displays maximum stability at intermediate pH values between the pI of leptin and CRH2; pH variations outside this range result in destabilization of the complex.

For the mouse and pig complexes, the interaction is favored under acidic conditions; however, as the pH approaches the pI of leptin, the complex becomes destabilized. Further destabilization occurs at pH values exceeding the pI of CRH2.

These differential behaviors are associated with the presence of pH-sensitive residues at the leptin–CRH2 interface, such as His78 in rat leptin and Arg5 in pig leptin, which undergo protonation–deprotonation transitions near physiological pH. Additionally, His439 (macaque), Arg466(468), Arg467 (mouse and rat), Asp530(532), Lys558 (macaque), Glu563(565), Arg613(615), and Asp615(617) residues in the CRH2 interface, which are also susceptible to protonation shifts, may also contribute to the observed variation in pH-dependent stability across species.

These findings are relevant considering local pH fluctuations modify protein–protein interactions by disrupting electrostatic contacts in different physiopathological conditions such as inflamed, hypoxic, or tumor tissues [41,42]. For instance, tumor tissues often exhibit extracellular pH values around 6.5 or lower [28,43], potentially reducing leptin–Ob-R binding efficiency and affecting downstream signaling.

Inter-tissue pH differences across species and conditions may also contribute to species-specific stability profiles, reinforcing the notion that leptin–Ob-R interaction is environmentally modulated [44,45].

#### 2.4.2. Effect of Ionic Strength on Binding Stability

We next examined the effect of ionic strength (I) on predicted ΔG_b_ over a physiological to supra-physiological range (0.05– 0.40 M). All complexes displayed increased stability with rising ionic strength, as evidenced by more negative predicted ΔG_b_ values (Figure 4B). The mouse complex was particularly sensitive, showing a ΔG_b_ improvement of nearly 2 kcal/mol. This stabilizing effect likely reflects enhanced shielding of repulsive electrostatic interactions and reinforcement of hydrophobic contacts at the binding interface [22]. Ionic strength is a critical modulator of protein interactions in biological systems. In normal tissues, extracellular ionic conditions are maintained through tightly regulated levels of Na^+^, K^+^, Cl^−^, Ca^2+^, and Mg^2+^ [23,46,47]. However, pathological conditions such as chronic inflammation, cancer, and ischemia may disrupt ion homeostasis, altering the electrostatic landscape around cell surface receptors [22], including Ob-R.

The observed increase in binding stability suggests that the leptin–CRH2 interaction is favored in ion-rich microenvironments, which could occur in highly vascularized tissues or during acute immune responses. Moreover, tissue-specific ion channel activity and ionic gradients (e.g., in neurons, immune synapses, or tumor stroma) may dynamically influence leptin signaling through modulated receptor binding affinity.

#### 2.4.3. Effect of Temperature on Binding Stability

To assess thermal sensitivity, leptin–CRH2 complex stability in the species was analyzed across a temperature range of 288 to 323 K (15 to 50 °C). In all five species, increasing temperature led to a decrease in binding affinity (ΔG_b_ became less negative by ~1 kcal/mol), indicating destabilization of the complex models (Figure 5A). This behavior reflects the weakening of noncovalent forces, particularly electrostatic and hydrophobic interactions, which maintain the heterodimer conformation [48].

Van ’t Hoff plots (Figure 5B) revealed positive slopes across all models, confirming that complex dissociation is enthalpically driven and that the formation of the complex is exothermic. The mouse complex, which exhibited the most hydrophobic interface, also showed the steepest decline in binding affinity with temperature, consistent with greater temperature sensitivity of hydrophobic contacts.

Although mammalian core body temperature is tightly regulated (~37 °C), localized variations occur under physiological and pathological states. For example, elevated tissue temperature has been documented in breast tumors [49,50] and inflammatory foci, potentially affecting receptor–ligand binding kinetics. Leptin is implicated in cancer cell proliferation, immune evasion, and inflammation [51], raising the possibility that thermally induced dissociation of the leptin–CRH2 complex may contribute to altered signaling under febrile or tumor-associated conditions.

**Figure 5 ijms-26-06955-f005:**
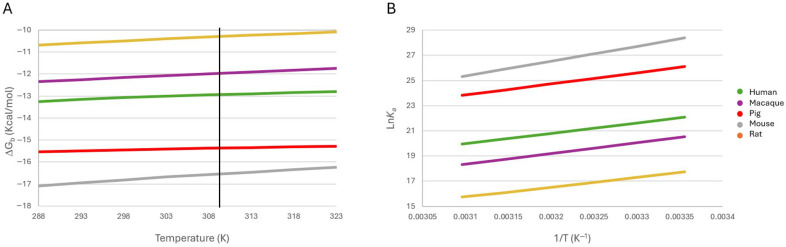
Effect of temperature changes on the stability (**A**) and *K_a_* (**B**) of leptin–CRH2 heterodimer models of five mammalian species. The vertical line in (**A**) indicates the physiological temperature. The data were predicted from FoldX.

Overall, our findings demonstrate that pH, ionic strength, and temperature are significant modulators of leptin–CRH2 complex stability, with distinct effects across species. These parameters must be considered when modeling receptor–ligand interactions in variable tissue environments or disease contexts.

### 2.5. Physiological and Pathological Implications

The leptin–Ob-R axis plays an integral role in coordinating metabolic and immune responses. Given the expression of both leptin and its receptor across a wide range of tissues, including adipose tissue, the hypothalamus, mammary glands, skeletal muscle, immune cells, and various tumors, the modulation of leptin binding by physicochemical conditions is likely to be functionally relevant in vivo.

Our data suggest that acidic microenvironments, such as those found in tumors or at sites of chronic inflammation, may stabilize/destabilize, based on each species, the leptin–CRH2 complex models and modulate downstream signaling pathways. For instance, in breast cancer, leptin promotes tumor cell proliferation, migration, and immune evasion, particularly under obesity-associated low-pH conditions [52,53]. Similarly, elevated extracellular ion concentrations and inflammatory cytokine release during immune activation could stabilize the leptin–Ob-R interaction, enhancing pro-inflammatory signaling [2,54].

Temperature increases, whether due to fever, local inflammation, or thermogenic activity in brown adipose tissue [55], may also contribute to reduced binding stability. In many cancerous tissues, where Ob-R expression is upregulated [56], localized hyperthermia could transiently weaken receptor–ligand interactions, potentially altering signaling outcomes. These findings highlight the context-dependent nature of leptin activity and support the concept of the microenvironmental regulation of cytokine-like hormones.

Furthermore, interspecies differences observed in binding energy and residue contributions provide a molecular basis for variability in leptin responsiveness across model organisms. This highlights the importance of considering species-specific receptor architectures and local tissue conditions when extrapolating experimental findings to human physiology or developing therapeutic leptin analogues.

### 2.6. Thermodynamic Analysis

The Van ‘t Hoff plots presented in Figure 5B show the effect of temperature change on the binding constants. They show the correlation between Ln*K_a_* and the inverse of the temperatures of the five leptin–CHR2 dimers. The five complexes show positive slopes. This means that, when the temperature increases, the predicted *K_a_* of the models decreases, favoring the dissociation of complexes. This also means that complex formation is an exothermic reaction.

Regarding the contribution profiles of ΔH and ΔS of the different leptin–CRH2 models (Table 1), we observe that the mouse leptin–CRH2 complex has the highest enthalpic and entropic contributions among all species. The pig leptin–CRH2 complex has the second highest enthalpic contribution, followed by the macaque, human, and rat leptin–CRH2 complexes. Regarding the entropic construction, the rat has the second largest contribution, followed by the macaque, human, and pig. Additionally, our findings indicate that the stability and binding energies of leptin–CRH2 complexes under these physicochemical conditions are driven mainly by enthalpic contributions. These results support a model in which subtle sequence variations modulate binding via both direct contacts and overall interface architecture.

This strategy can be applied to visualize the differences in interactions that occur at the interface of heterodimers and homodimers of different species, to predict the impact of stability and affinity between two proteins, and to project the thermodynamics and stability behavior of any complex under the effect of different microenvironmental conditions that are comparable to physiological or pathophysiological conditions. The application of this strategy will make it possible to address molecular, energetic, adaptive, and thermodynamic aspects between two proteins in different tissues.

## 3. Materials and Methods

### 3.1. Selection and Preparation of Homologous Sequences

Leptin and CRH2 domain sequences were obtained from the UniProt Knowledgebase ((https://www.uniprot.org/uniprotkb) accessed on 21 March 2025) [57]. Only reviewed entry sequences were used in this work. Five biological species met this criterion and were analyzed: human, mouse, rat, pig, and macaque. The accession numbers for leptin are P41159 (human), P41160 (mouse), Q29406 (pig), Q28504 (macaque), and P50596 (rat). The leptin sequences analyzed correspond to the mature proteins after removal of the signal peptide (typically 21 amino acids), resulting in 146 residues for leptin. The accession numbers for CRH2 of Ob-R used are P48357 (human), P48356 (mouse), O02671 (pig), Q9MYL0 (macaque), and Q62959 (rat). Sequences of mature leptin and corresponding CRH2 domains were aligned using Clustal Omega (v1.2.4) [58]. To avoid any confusion, the numbering of CRH2 sequences is given based on each of the biological species, and the position of the same residue in CRH2 in humans is shown in parentheses or otherwise stated.

### 3.2. Model Construction and Structural Optimization

Although a crystallographic structure of the leptin–CRH2 heterodimer has recently been published [33] (PDB 7Z3Q), its direct use as a modeling template presented some limitations. The available coordinates are missing 44 residues from leptin (out of 146) and six residues from the CRH2 domain of Ob-R (out of 202), which include regions that may impact structural integrity and interface definition. Two key residues (N516Q and C604S) in the CRH2 domain are substituted in the sequence of the crystallographic construct, potentially altering native interaction patterns. Given these discrepancies and the importance of full-length and sequence-accurate models for reliable interface analysis, analysis of the dependence of stability on the effect of pH and ionic strength, and thermodynamic analysis, we instead employed a homology-based docking approach, followed by molecular dynamics simulation (MDS) refinement, to generate complete and sequence-validated models for each species analyzed.

Homology models of the leptin–CRH2 complex were constructed for each species using the SWISS-MODEL server ((https://swissmodel.expasy.org/) accessed on 25 March 2025) employing the Protein–Complex Molecular Dynamics Model (PCMDM) template. Briefly, the 3D models of leptin and CRH2 were constructed using the I-Tasser server ((https://zhanggroup.org/I-TASSER/) accessed on 16 July 2025), employing the coordinates of crystallographic structures 1AX8 and 3V6O, respectively. The leptin-CRH2 complex model was obtained by docking on the GRAMM-X server ((https://gramm.compbio.ku.edu/) accessed on 16 July 2025). Subsequently, the complex was subjected to molecular dynamics simulations (MDS) with the GROMACS program ((https://www.gromacs.org/) accessed on 16 July 2025) for 100 ns at 310 K and with the OPLS-AA force field. The PCMDM template refers to the average structure of the most representative cluster derived from MDS of the docked human leptin–CRH2 complex [34]. This stable model was used as a comparative structural reference across species. Model quality was evaluated using QMEANDisCo scores [59], the PDBefold Q-scores [60,61], and root mean square deviation (RMSD) of backbone atoms. The molecular graphics, modeling, and simulation program YASARA ((Available online: https://www.yasara.org/) accessed on 30 March 2025) was used for energy minimization and structural visualization, and for comparison [62,63].

### 3.3. Interface Characterization

The molecular interface between leptin and CRH2 was analyzed using PDBePISA ((https://www.ebi.ac.uk/pdbe/pisa/) accessed on 1 April 2025) [64] to determine the solvent-accessible surface area (SASA), identify interface residues, and classify interaction types (e.g., hydrogen bonds, salt bridges, hydrophobic contacts). Interaction surfaces were visualized, and contact residues were compared across species.

For a further exploration of the interaction interface of the five leptin–CRH2 complexes, we used PRODIGY ((https://rascar.science.uu.nl/prodigy/) accessed on 5 April 2025) [37,38], which has a 5.5 Å cutoff distance. The temperature was set at 36.5 °C.

### 3.4. Binding Energy Prediction and Thermodynamic Analysis

The change in binding free energy (ΔG_b_) predicted for each leptin–CRH2 complex was estimated using the FoldX software ((Available online: https://foldxsuite.crg.eu/) accessed on 8 April 2025) [65,66], which employs an empirical force field based on data from two databases of fragments of crystallized proteins: BackXDB and LoopXDB. The Repair command was used for energy minimization, and the AnalyseComplex command was employed to evaluate heterodimer binding in terms of ΔG_b_ of each biological species. Predictions were made under standard physiological conditions (pH 7.0, 0.15 M ionic strength, 298 K) and then repeated across variable conditions to assess sensitivity to pH (3.0–10.0), ionic strength (0.05–4.0 M), and temperature (288–323 K).

Predicted equilibrium binding constants (*K_a_*) were calculated from predicted ΔG_b_ values using the following well-established thermodynamic relationship:(1)$$∆G_{b}=-RT\ln K_{a}$$
and with these values, the well-known Van ‘t Hoff equation was generated to determine the affinity(2)$$\ln K_{a}=\left(-\frac{\Delta H}{R}\right)\left(\frac{1}{T}\right)+\left(\frac{\Delta S}{R}\right)$$

The predicted values of enthalpy (Δ*H*) and entropy (Δ*S*) changes associated with the affinity of the leptin–CRH2 complex were calculated from the slope (m = $-\frac{\Delta H}{R}$) and the intercept (*b* = $\frac{\Delta S}{R}$), respectively, as reported elsewhere [40]. This allowed the decomposition of energetic contributions governing complex stability across species.

## 4. Conclusions 

This study presents a comparative structural and thermodynamic characterization of the leptin–CRH2 interaction in five mammalian species using experimentally validated sequences and homology modeling. Despite the overall conservation of the binding interface, species-specific sequence variations influence complex stability, contact architecture, and sensitivity to physicochemical conditions.

Our findings reveal that the interaction is predominantly enthalpy-driven, with noncovalent interactions, such as hydrogen bonds and salt bridges, and hydrophobic interactions playing a significant role in complex stabilization. Acidic environments and elevated temperatures weaken binding affinity; conversely, high ionic strength enhances stability across all models.

These results contribute to our understanding of how leptin signaling is modulated in response to microenvironmental stress. By mapping residue-level contributions to binding energetics, this study enables predictive modeling of how naturally occurring or engineered substitutions may affect leptin–Ob-R interactions under diverse physiological and pathological conditions. This work supports the development of context-sensitive leptin modulators and improves the translational relevance of animal models in immunometabolic and inflammatory research.

## Figures and Tables

**Figure 1 ijms-26-06955-f001:**
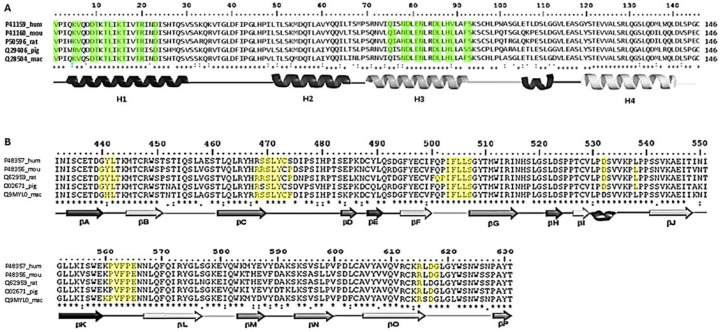
Multiple sequence alignment of five mature leptin homologs (**A**) and five homologs of the CRH2 domain of Ob-R (**B**). The secondary structures of leptin and CRH2 are shown below each multiple alignment. The residues involved in the interface of the models are highlighted in green and yellow for leptin and CRH2, respectively. In this figure, the positions of residues are given according to the numbering of leptin and CRH2 in humans. Asterisk indicates positions which have a single, fully conserved residue; colon indicates conservation between groups of strongly similar properties; and period indicates conservation between groups of weakly similar properties.

**Figure 2 ijms-26-06955-f002:**
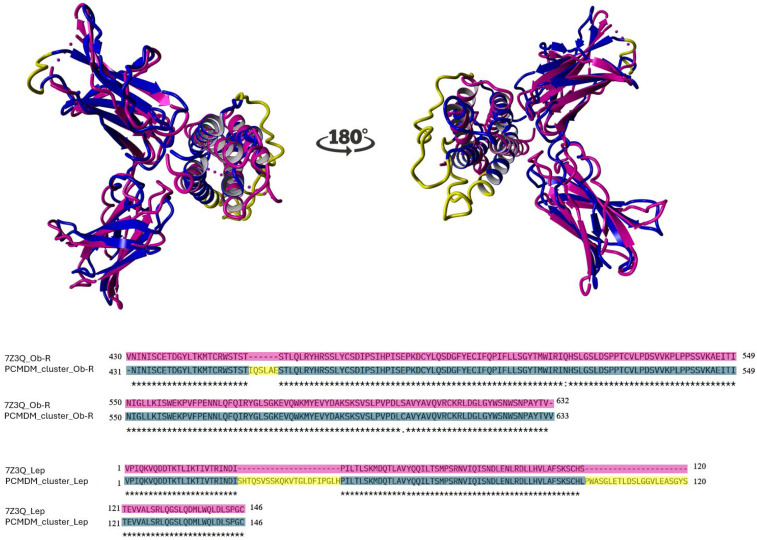
Structural and sequence alignment of the human leptin–CRH2 complex. The crystallographic structure from PDB 7Z3Q is shown in magenta, while the complete, full-length PCMDM is shown in blue and yellow. The image displays two 180° opposite views of the aligned heterodimers. Residues absent in the 7Z3Q coordinates but modeled in the PCMDM structure are highlighted in yellow, both in the 3D structures and in the corresponding sequence alignments below. Discontinuities (gaps) in the 7Z3Q structure are marked by magenta spheres. These missing regions, which include 44 residues from leptin and 6 from CRH2, were reconstructed through homology modeling followed by molecular dynamics simulation to ensure full-length, sequence-validated models for downstream analysis.

**Figure 3 ijms-26-06955-f003:**
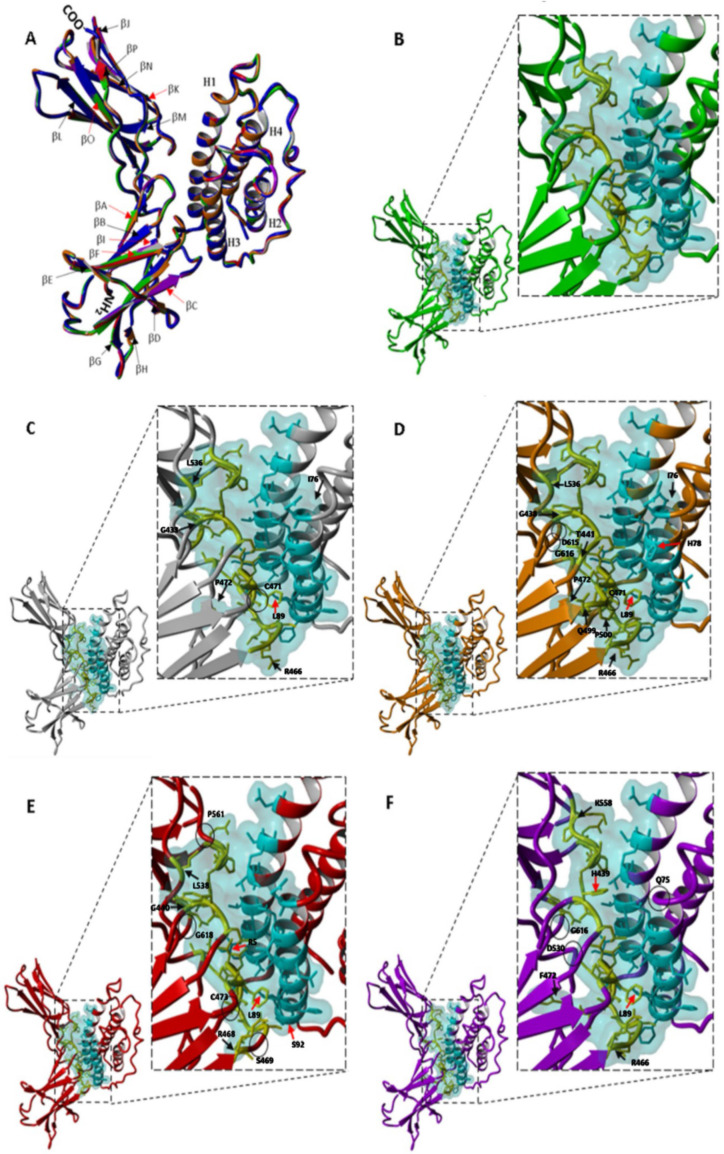
Structural alignment and comparative visualization of the leptin–CRH2 complex interfaces across species. (**A**) Structural superposition of the alpha-carbons of leptin–CRH2 complexes from five species: human (green), mouse (gray), rat (orange), pig (red), and macaque (purple), aligned using the PCMDM template (blue). (**B**–**F**) Interface views of the leptin–CRH2 heterodimers for human, mouse, rat, pig, and macaque, respectively. The interface contact surface is shown in cyan. Participating residues from leptin are colored cyan, and those from CRH2 are in yellow and shown as sticks. Residues that are not involved in the human interface but contribute to the interaction in other species are indicated by black arrows. Substituted amino acids that still contribute to the interface are indicated by red arrows. Residues highlighted by black circles participate in the human interface but not in the other species. Residue numbering corresponds to the specific protein sequence of each species.

**Table 1 ijms-26-06955-t001:** Predicted binding free energy and thermodynamic properties of leptin–CRH2 complexes from five mammalian species.

Species	*ΔG_b_ (kcal/mol)	**K_d_* M (× 10^−10^)	**K_a_* M^−1^(× 10^10^)	^$^ΔH (kcal/mol)	^$^-TΔS (kcal/mol)
Human	−13.17	2.54	0.392	−16.28	3.11
Mouse	−16.81	4612.39	216.807	−23.45	6.64
Rat	−10.50	0.01	0.005	−15.36	4.87
Pig	−15.44	462.98	21.599	−17.31	1.87
Macaque	−12.152	0.12	0.082	−16.93	4.78

* These data were predicted using the FoldX server at 298 K, pH 7.0, and ionic strength (I) of 0.15 M. *K_a_* is the association constant, and *K_d_* is the dissociation constant. ^$^ These data were calculated as mentioned in the Section 3.4 on thermodynamic analysis.

**Table 2 ijms-26-06955-t002:** Effects of temperature, pH, and ionic strength on the predicted *K_a_* of leptin–CRH2 complex models of 5 mammalian species.

Human	Temperature (°C)	*K_a_* [M]^*^	pH	*K_a_* [M]^#^	I [M]	*K_a_* [M]^$^
	25	3.93 × 10^9^	3	3.02 × 10^9^	0.05	4.09 × 10^9^
	30	2.46 × 10^9^	4	3.12 × 10^9^	0.1	3.83 × 10^9^
	35	1.57 × 10^9^	5	3.27 × 10^9^	0.15	3.93 × 10^9^
	40	1.03 × 10^9^	6	3.25 × 10^9^	0.2	4.81 × 10^9^
	45	6.86 × 10^8^	7	3.93 × 10^9^	0.25	5.80 × 10^9^
			8	3.93 × 10^9^	0.3	6.91 × 10^9^
			9	3.93 × 10^9^	0.35	7.84 × 10^9^
			10	3.93 × 10^9^	0.4	8.69 × 10^9^
Mouse	Temperature (°C)	*K_a_* [M]*	pH	*K_a_* [M]^#^	I [M]	*K_a_* [M]^$^
	25	2.17 × 10^12^	3	2.33 × 10^12^	0.05	7.69 × 10^11^
	30	1.09 × 10^12^	4	2.33 × 10^12^	0.1	1.36 × 10^12^
	35	5.76 × 10^11^	5	2.17 × 10^12^	0.15	2.17 × 10^12^
	40	3.16 × 10^11^	6	2.17 × 10^12^	0.2	3.02 × 10^12^
	45	1.76 × 10^11^	7	2.17 × 10^12^	0.25	4.08 × 10^12^
			8	2.17 × 10^12^	0.3	5.31 × 10^12^
			9	1.97 × 10^12^	0.35	6.81 × 10^12^
			10	1.97 × 10^12^	0.4	8.66 × 10^12^
Rat	Temperature (°C)	*K_a_* [M]*	pH	*K_a_* [M]^#^	I [M]	*K_a_* [M]^$^
	25	5.09 × 10^7^	3	4.71 × 10^7^	0.05	5.13 × 10^7^
	30	3.25 × 10^7^	4	4.71 × 10^7^	0.1	4.80 × 10^7^
	35	2.11 × 10^7^	5	4.71 × 10^7^	0.15	5.09 × 10^7^
	40	1.42 × 10^7^	6	4.71 × 10^7^	0.2	5.53 × 10^7^
	45	9.74 × 10^6^	7	5.09 × 10^7^	0.25	6.10 × 10^7^
			8	5.09 × 10^7^	0.3	6.82 × 10^7^
			9	4.72 × 10^7^	0.35	8.02 × 10^7^
			10	4.72 × 10^7^	0.4	9.42 × 10^7^
Pig	Temperature (°C)	*K_a_* [M]*	pH	*K_a_* [M]^#^	I [M]	*K_a_* [M]^$^
	25	2.16 × 10^11^	3	2.16 × 10^11^	0.05	1.54 × 10^11^
	30	1.31 × 10^11^	4	2.16 × 10^11^	0.1	1.85 × 10^11^
	35	8.29 × 10^10^	5	2.09 × 10^11^	0.15	2.16 × 10^11^
	40	5.23 × 10^10^	6	2.09 × 10^11^	0.2	2.51 × 10^11^
	45	3.37 × 10^10^	7	2.16 × 10^11^	0.25	2.74 × 10^11^
			8	1.99 × 10^11^	0.3	2.99 × 10^11^
			9	1.99 × 10^11^	0.35	3.32 × 10^11^
			10	1.99 × 10^11^	0.4	3.83 × 10^11^
Macaque	Temperature (°C)	*K_a_* [M]*	pH	*K_a_* [M]^#^	I [M]	*K_a_* [M]^$^
	25	8.28 × 10^8^	3	5.25 × 10^8^	0.05	7.07 × 10^8^
	30	5.13 × 10^8^	4	5.25 × 10^8^	0.1	7.34 × 10^8^
	35	3.24 × 10^8^	5	5.25 × 10^8^	0.15	8.28 × 10^8^
	40	2.08 × 10^8^	6	7.03 × 10^8^	0.2	9.45 × 10^8^
	45	1.35 × 10^8^	7	8.28 × 10^8^	0.25	1.02 × 10^9^
			8	8.33 × 10^8^	0.3	1.11 × 10^9^
			9	7.66 × 10^8^	0.35	1.25 × 10^9^
			10	7.66 × 10^8^	0.4	1.42 × 10^9^

* These data were calculated from the predicted ΔG_b_ obtained in FoldX, keeping the pH at 7.0 and 0.15 M ionic strength. ^#^ These data were calculated from the predicted ΔG_b_ obtained in FoldX, keeping the temperature at 25 °C and 0.15 M ionic strength. ^$^ These data were calculated from the predicted ΔG_b_ obtained in FoldX, keeping the temperature at 25 °C and pH at 7.0.

## Data Availability

The data presented in this study are available upon request from the corresponding author.

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
