# Peer review of "Bioinformatic Analysis of the Leptin–Ob-R Interface: Structural Modeling, Thermodynamic Profiling, and Stability in Diverse Microenvironments"

_ijms, 2025, doi:10.3390/ijms26146955_

Round 1
Reviewer 1 Report
Comments and Suggestions for Authors
In this study, the authors construct five models of leptin bound to the CRH2 domain of the leptin receptor (leptin-CRH2) for human, mouse, rat, pig, and macaque species. They then analyze and compare the interfaces of each heterodimer and predict the binding free energy (ΔGb) and related thermodynamic properties under standard conditions. Finally, they extend their analysis to predict how variations in temperature, pH, and ionic strength affect ΔGb. For this work, the authors relied on several tools and databases available online.
Main issues:
The manuscript is difficult to follow because it often lacks sufficient context. Although the Methods section is placed after the Results and Discussion, the Results should be presented clearly enough for readers to understand what was done without needing to refer immediately to the detailed methodology.
For example, lines 94-95 refers to a PCMDM template. What is this template? It’s not even described in the Methods section. Is it the full-length human lectin-CRH2 complex? Where does it come from? If it is the human full-length complex (shown in blue in figure 3), then what’s the difference between this template and the human complex produced by the authors (shown in green in figure 3)?
This whole paragraph (lines 90-106) is very confusing. Why is the RMSD computed using only 186 or 183 residues? The 7Z3Q experimental structure of leptin and Ob-R has 102 and 197 visible residues, respectively. Why not use all of them? And why two different calculations (with 186 and 183 aa)?
In Figure 2, the discontinuities in the amino acid sequence of leptin and its receptor in the 7ZQ3 model are represented as small magenta spheres. In blue, which should show the complete, full-length structure, the modeled amino acids that are absent in the 7ZQ3 model should appear in these regions but are not. Why? Are the authors confident that they have constructed a complete, full-length model?
The interface analysis (Section 2.2) should also be performed on the experimentally determined 7Z3Q human complex and compared to the simulated human complex generated by the authors to assess whether it is close to the experimental structure.
In Section 2.3 (Stability Analysis of Leptin-CRH2 Complexes), the authors present predicted Kd values for each complex and state that, for the human species, the predicted value is very similar to the previously reported experimental value. However, the actual experimental value is not provided - what is it? Additionally, for the mouse complex, the previously reported experimental data (0.1 to 13 nM) differs substantially from the predicted value (~460 nM). This rather large difference is worrying, because if it exists for mouse, then what guarantees us that the values for the pig, rat and macaque complexes are close to the real ones?
Section 3.2 (Model Construction and Structural Optimization) is poorly described. It is not clear what was done in this work and what comes from previous work. A reference is made to refinement by molecular dynamics simulations, but no details are given. I would like to remind the authors that the methods should be detailed enough so that others can reproduce the work. If the work uses structures derived from previous work, then the models used as a starting point could be provided as supplementary materials. The Model Construction was made with the SWISS-MODEL server, but no details are given, as well as no details regarding the Structural Optimization.
Since this manuscript deals with three-dimensional protein structures, the introduction should have a brief description of the structure of leptin and its receptor (and the CRH2 domain in particular).
Finally, the references should be carefully reviewed throughout the manuscript, as there are several instances where statements cite articles that are not actually relevant to the content being discussed. A list of some of the issues identified is provided below, (Reference issues).
Minor issues
Line 21: “greater clarity in structural and stability analysis”: the gain in clarity should be on the structural and stability of the complex, and not on the analysis.
Line 24: “Binding free energy” should be “Predicted binding free energy?”
Lines 37-40: For the sake of clarity, the tissues where lectin is expressed should be separated from the tissues where the receptor is expressed.
Line 74: Remove “The text continues here”
Figure 1: What are the red squares in some residues of human leptin?
Line 91: Please insert here a reference to the structure that you are referring to.
Line 92: “two substitutions in key receptor positions”. What are these residues? Why are they in “key positions”?
Line 94: What is PCDMD?
Lines 95, 113: Z7Q3 is not in the PDB. Did you mean 7Z3Q?
Line 122: What is “Le Interface analysis?
Line 141: change a-carbons to alpha-carbons. Please clarify whether in A five different models are shown.
Line 166: What do you mean with “their exploration ”
Line 168: What is MDS?
Line 201: “At 36.5 °C“ the temperature seems irrelevant for the computation of the number of interactions.
Line 202: What do you mean with “These last explorations“?
Line 210: 0.15M of what? Is it ionic strength
Line 216-217, Table 2: Please define Kd and Ka; Footnote: Please define I on first use.
Line 239: “The Ka “ should be “The predicted Ka “
Figure 4: I don't see the point in using the abbreviations h, mac, pig, mou and rat, followed by WT, to refer to the different species. Why not be consistent and use the same nomenclature as in Table 3?
Figure 5: See comments for figure 4 above.
Line 378: 3.1. “Thermodinamics Analysis”: The title of this sub-section is clearly wrong.
Lines 382-383: The UniProt leptin sequences for the five species have 167 aa. However, in Figure 1, leptin sequences have 146 aa. Please describe any existing post-translational modifications. The same could be said for the leptin receptor
Line 395: What are these Two key residues ?
Section 3.3: Is there any difference between ΔG (line 420) and ΔGb (lines 424 and 430)?
References issues
Line 40: Please check reference 4 (Hübner, 2012) because it does not seem to relate to expression of leptin and Ob-R…
Line 89: Please check reference 28, because it is not related to residue conservation of leptin or Ob-R.
Line 103: Please check reference 29, because it seems to only describe a tool to calculate RMSD values.
Line 117: I don’t see the relation of references 28 and 31 to the text.
Line 293: Reference 43 deals with thermal stability and not ionic strength. Please insert a suitable reference.
Line 386: Reference 54 is about the “EMBL-EBI Job Dispatcher” service. Although the authors cite (and quite rightly so) this reference for using the online service, in my opinion they should also cite the original reference for the Clustal Omega program (Sievers. and Higgins, 2021)
Comments on the Quality of English LanguageThe language needs to be improved. Here is a non-exhaustive list of the issues found:
Lines 14-15: It is not clear to me what the authors mean by “While the first interaction between leptin–Ob-R structural aspects have been studied in humans and mice…”, but perhaps it should be rephrased as “While the structural aspects ofthe interaction between leptin–Ob-R have been first studied in humans and mice…”
Line 18: Change "leptin–CRH2" to "leptin–CRH2 complex"
Line 19: Change "new template of full-length" to "full-length template"
Line 23: its --> is
Line 42: have identified --> were identified
Line 66: “, characterized” --> “and characterize”
Line 68 mimic --> mimicking?
Line 85: Please rephrase: “The alignment CRH2 of five Ob-R homologs“
Line 95: leptinCRH2 --> leptin–CRH2
Line 125: Serin --> serine.
Line 172: Perhaps “differences” is a better word than “changes”?
Line 186: Please change “, as template in combination of literature” to “as template, in combination with literature”
Line 200: "In table 1" --> "Table 1"
Line 216: The word “homologous” is misplaced.
Lines 253-254: Please rephrase.
Author Response
MDPI Manuscript ID: ijms-3715096
Title: Bioinformatic analysis of the leptin–Ob-R interface: structural modeling, thermodynamic profiling, and stability in diverse microenvironments
Reviewer 1
On behalf of the authors, I would like to thank you for the constructive feedback and thoughtful evaluation of our manuscript. We appreciate the time and expertise invested in the review process because all your suggestions help us importantly to improve the clarity of the manuscript. Below, we provide a point-by-point response to each of the comments and suggestions, along with a description of the changes made. All changes in the manuscript are written in red.
Thank you in advance for your kind attention to my request.
Reviewer 1
In this study, the authors construct five models of leptin bound to the CRH2 domain of the leptin receptor (leptin-CRH2) for human, mouse, rat, pig, and macaque species. They then analyze and compare the interfaces of each heterodimer and predict the binding free energy (ΔGb) and related thermodynamic properties under standard conditions. Finally, they extend their analysis to predict how variations in temperature, pH, and ionic strength affect ΔGb. For this work, the authors relied on several tools and databases available online.
Main issues:
We would like to point out that the addition of new sentences in the revised version has altered the original sentence numbering. To enhance traceability and facilitate reviewer assessment, when the numbering changed, we have included both the original line numbers from the submitted manuscript and the corresponding updated line numbers from the revised version. Updated line numbers appear in parentheses immediately after the original ones (e.g., “Line 91 (now Line 104)”) throughout our responses.
- The manuscript is difficult to follow because it often lacks sufficient context. Although the Methods section is placed after the Results and Discussion, the Results should be presented clearly enough for readers to understand what was done without needing to refer immediately to the detailed methodology.
ANSWER
We appreciate this observation and agree that results should be as self-contained as possible. To address this, we added sentences in the Results section to include a brief methodological context at the beginning of each subsection. For example, we now explicitly mention the tools used (e.g., Clustal Omega for sequence alignment, YASARA for molecular dynamics, etc.) when first referring to each analysis. The sentences were added at the beginning of the following sections:
2.1. Sequence Conservation and Structural Consistency
“To investigate evolutionary conservation and suitability for structural modelling, we first compiled revised leptin and CRH2 sequences from five species (human, mouse, rat, pig, and macaque), from UniProt and them using Clustal Omega. We then constructed homology models for the leptin–CRH2 complexes from five species using validated templates.”
2.2. Interface Variation Across Species
“Following model generation and refinement through molecular dynamics simulations (MDS), we analyzed the interface residues using PDBePISA and visualized contact patterns using YASARA. Comparisons among species highlighted conserved and variable interaction zones.”
2.3. Analysis of the Stability of Leptin–CRH2 Complexes
“We calculated the predicted free binding energy (ΔGb) using the FoldX server under physiological conditions (pH 7.0, 298 K, 0.15 M ionic strength).”
2.4. Effects of Physicochemical Conditions on Leptin–CRH2 Complex Stability
“We next simulated how changes in microenvironmental conditions (pH, ionic strength, and temperature) affect the thermodynamic stability of leptin–CRH2 complexes across species. These conditions were selected to reflect physiological and pathological tissue variation.”
We think it might improve the clarity and accessibility of the manuscript and help the reader follow the logic and design of each result without needing to refer immediately to the detailed methodology.
- For example, lines 94-95 (112-114) refers to a PCMDM template. What is this template? It’s not even described in the Methods section. Is it the full-length human leptin-CRH2 complex? Where does it come from? If it is the human full-length complex (shown in blue in figure 3), then what’s the difference between this template and the human complex produced by the authors (shown in green in figure 3)?
ANSWER
Thank you for pointing out the lack of clarity regarding the term “PCMDM template.” We agree that this term was not properly defined in the manuscript. The PCMDM template refers to a representative structure obtained from the Protein–Complex Molecular Dynamics Modeling process reported by Lopez-Hidalgo et al [34]. It corresponds to the averaged structure of the largest cluster resulting from MD simulations of the docked human leptin–CRH2 complex. This structure served as a stable reference for aligning and evaluating all species-specific models.
The PCMDM coordinates (Table S1) were used as template to obtain human leptin–CRH2 complex shown in Figure 3, using Swiss Model, as it was made for the rest of the models of the biological species analyzed in this work.
The manuscript has been corrected in the Methods and Results sections to clarify this statement.
- This whole paragraph (lines 90-106) (now 104-129) is very confusing. Why is the RMSD computed using only 186 or 183 residues? The 7Z3Q experimental structure of leptin and Ob-R has 102 and 197 visible residues, respectively. Why not use all of them? And why two different calculations (with 186 and 183 aa)?
ANSWER
Thank you for highlighting this issue. We acknowledge that the paragraph was confusing and insufficiently explained. The number of residues used in the RMSD calculations (183 and 186) reflects the number of aligned residues between the PCMDM model and the crystallographic structure (PDB: 7Z3Q) that could be matched with confidence during superposition. We did not use all 102 leptin and 197 CRH2 residues from 7Z3Q because some were either absent in the model template (due to disorder or truncation) or excluded by the alignment algorithm due to significant deviations or lack of one-to-one correspondence.
The difference between 183 and 186 residues stems from using two different structural comparison tools (YASARA and PDBeFold) which apply slightly distinct alignment criteria and gap-handling rules. We have modified the text trying to add clarity to these statements in the corresponding sentences at the Results section.
The new paragraph now it is as follows:
“Due to these limitations, we opted to generate full-length, sequence-validated models for each species via homology modeling using as template the coordinates of PCMDM model as template, which corresponds to the average structure of the most representative cluster obtained from molecular dynamics simulations of the human leptin–CRH2 complex previously reported [32]. This model was used as a structural reference for alignment and stability comparison across all species. To evaluate the similarity between the PCMDM model and the 7Z3Q crystallographic structure, we performed pairwise 3D alignments. Using YASARA, we obtained an RMSD of 2.083 Å based on 186 aligned residues with 93.01% sequence identity. Using PDBeFold, the RMSD was 2.304 Å over 183 aligned residues with 85% sequence identity. The difference in aligned residue counts reflects variations in the algorithm’s handling of mismatches and gaps. Importantly, not all residues from the crystal structure were used because some segments were missing or poorly resolved, and some interface regions could not be reliably matched. The observed RMSD differences were influenced by specific substitutions of Asparagine516 for Glutamine and Cysteine604 for Serine and the absence of residues 454-IQSLAE-459 in the Ob-R coordinates, as well as by the absence of residues 25-SHTQSVSSKQKVTGLDFIPGLH-46 and 99-PWASGLETLDSLGGVLEASGYS-120 in the coordinates of leptin in the 7ZQ3 structure. Structural similarity, assessed by RMSD, is a widely accepted measure in homology-based protein modeling, as it quantifies atomic superposition fidelity and helps evaluate model quality.”
- In Figure 2, the discontinuities in the amino acid sequence of leptin and its receptor in the 7Z3Q model are represented as small magenta spheres. In blue, which should show the complete, full-length structure, the modeled amino acids that are absent in the 7ZQ3 model should appear in these regions but are not. Why? Are the authors confident that they have constructed a complete, full-length model?
ANSWER
Thank you for the observation. We agree that the initial figure and accompanying text did not sufficiently clarify the location and visualization of the missing residues from the 7Z3Q crystal structure. In the revised version, we have updated Figure 2 to clearly highlight in yellow the amino acids that are absent in the 7Z3Q structure but present in the PCMDM full-length models, both in the 3D alignment and in the sequence alignment below. These residues correspond to segments that were omitted in the experimental coordinates due to crystallographic limitations.
The magenta spheres still indicate the breakpoints in the 7Z3Q structure, while the yellow-highlighted residues now explicitly show the modeled regions that fill those gaps in the PCMDM models. We confirm that our models include complete, full-length sequences of leptin and CRH2, and these reconstructed regions are visible in the updated figure and were included in all subsequent analyses (RMSD, interface contacts, stability, and thermodynamics). Text has been revised accordingly for clarity.
- The interface analysis (Section 2.2) should also be performed on the experimentally determined 7Z3Q human complex and compared to the simulated human complex generated by the authors to assess whether it is close to the experimental structure.
ANSWER
We appreciate your suggestion to visualize the interface of the 7Z3Q crystal structure. We have written a new paragraph detailing the differences between the two models. The new paragraph now it is as follows (189-198):
The interface of the 7Z3Q crystallographic model also shows that alpha-helices 1 and 3 of leptin and the six loops (βA-βB, βC-βD, βF-βG, βI-βJ, βK-βL, and βO-βP) of Ob-R are involved, as in the PCMDM model. The interface area of the 7Z3Q model is longer (837.7 A2) than that of the PCMDM model (790.4 A2). The difference in area between the models is mainly due to the additional amino acids involved in Ob-R in the 7Z3Q model. For example, Thr443, Arg468, Asp475, Gln501, Pro502, Leu530, Asn566, Asn567, and Leu619 of Ob-R and Asp8, Arg71, and Ile74 of leptin do not participate in the PCMDM model but do participate in the 7Z3Q model. On the other hand, Ser469, Pro561, Pro564, and Gly618 of Ob-R and Thr10, His88, Phe92, and Ser93 of leptin do participate in the PCMDM model but not in the 7Z3Q model.
- In Section 2.3 (Stability Analysis of Leptin-CRH2 Complexes), the authors present predicted Kd values for each complex and state that, for the human species, the predicted value is very similar to the previously reported experimental value. However, the actual experimental value is not provided - what is it? Additionally, for the mouse complex, the previously reported experimental data (0.1 to 13 nM) differs substantially from the predicted value (~460 nM). This rather large difference is worrying, because if it exists for mouse, then what guarantees us that the values for the pig, rat and macaque complexes are close to the real ones?
ANSWER
We appreciate the reviewer’s observation regarding the comparison between predicted and experimental Kd values. In response, we have now explicitly included the reported experimental Kd for the human leptin–Ob-R interaction (approx. 1–15.4 nM [39]) in the Results section. This value is in good agreement with our predicted Kd for the human complex (13.4 nM), supporting the validity of our modeling approach.
We acknowledge the discrepancy observed for the mouse complex, where our model predicts a Kd of ~460 nM, while experimental values range from 0.1 to 17 nM depending on the assay and isoform tested [16, 39]. This difference likely reflects structural deviations in modeled loop regions and limitations of current scoring functions under variable environmental assumptions.
To address the reviewer’s comment about the remaining species (pig, rat, macaque), we point out that no experimental Kd values are currently available in the literature for those complexes. Thus, our predicted values should be interpreted in a comparative rather than absolute framework, highlighting relative stability trends rather than precise affinity estimates. This rationale is now clarified in the revised manuscript.
We have updated Section 2.3 of the Results to include these experimental comparisons and clearly state the intended interpretative scope of our predictions.
- Section 3.2 (Model Construction and Structural Optimization) is poorly described. It is not clear what was done in this work and what comes from previous work. A reference is made to refinement by molecular dynamics simulations, but no details are given. I would like to remind the authors that the methods should be detailed enough so that others can reproduce the work. If the work uses structures derived from previous work, then the models used as a starting point could be provided as supplementary materials. The Model Construction was made with the SWISS-MODEL server, but no details are given, as well as no details regarding the Structural Optimization.
ANSWER
We thank the reviewer for highlighting the need for clarity in Section 3.2. In response, we have significantly expanded the description of the model construction and refinement process. We now clearly distinguish which steps were performed in this study and which procedures follow previous work. Additionally, we provided the coordinates of PCMDM template as a supplementary material (Table S1). We have included details on sequence validation, homology modeling via SWISS-MODEL, and structure refinement using YASARA force field. These improvements aim to enhance reproducibility and transparency.
- Since this manuscript deals with three-dimensional protein structures, the introduction should have a brief description of the structure of leptin and its receptor (and the CRH2 domain in particular).
ANSWER
We appreciate the reviewer’s suggestion to include a brief structural overview of leptin and its receptor, particularly the CRH2 domain, in the Introduction. We have now added a brief description summarizing their known structural features and relevance to receptor binding and signaling. This addition intends to provide non-expert readers with the necessary structural context to better understand the subsequent modeling and analysis presented in this work.
- Finally, the references should be carefully reviewed throughout the manuscript, as there are several instances where statements cite articles that are not actually relevant to the content being discussed. A list of some of the issues identified is provided below, (Reference issues).
ANSWER
We appreciate the reviewer’s attention to the relevance of the citations. We have carefully revised all references cited in the manuscript and made the corresponding changes.
Minor issues
- Line 21 (22): “greater clarity in structural and stability analysis”: the gain in clarity should be on the structural and stability of the complex, and not on the analysis.
ANSWER
We have changed this sentence to express that the gain in clarity refers to the structural and stability features of the complex, not to the analysis itself.
The revised sentence now reads:
“...to enhance clarity in the structural features and stability of the complex”
- Line 24: “Binding free energy” should be “Predicted binding free energy?”
ANSWER
We have made the correction throughout the document, to reflect that the values are computationally estimated.
- Lines 37-40 (37-39): For the sake of clarity, the tissues where leptin is expressed should be separated from the tissues where the receptor is expressed.
ANSWER
We revised this section to distinguish clearly between tissues expressing leptin and those expressing Ob-R.
The updated sentences is as follows:
Leptin is expressed in adipose tissue, placenta, and skeletal muscle, among others, while Ob-R is highly expressed in the hypothalamus, gastrointestinal tract, heart, liver, small intestine, prostate, ovary, lung, kidney and skeletal muscle.
- Line 74 (now line 82) : Remove “The text continues here”
ANSWER
This phrase has been eliminated.
- Figure 1: What are the red squares in some residues of human leptin?
ANSWER
The figure has been updated and no longer includes any red squares.
- Line 91 (now line 107): Please insert here a reference to the structure that you are referring to.
ANSWER
We added a citation to the 7Z3Q crystal structure: [33]
- Line 92 (now line 107): “two substitutions in key receptor positions”. What are these residues? Why are they in “key positions”?
ANSWER
We now specify that the substitutions are N516Q and C604S, located near the leptin-binding interface, which could alter electrostatic and hydrogen-bonding networks.
- Line 94 (now line 111): What is PCDMD?
ANSWER
PCDMD is a typo, which has been corrected to PCMDM. Also, the description of PCMDM has been added in both Methods and Results, the following sentence has been added to the results section:
……. “which corresponds to the average structure of the most representative cluster obtained from molecular dynamics simulations of the human leptin–CRH2 complex previously reported [34].”
- Line 95, (now line 116): Z7Q3 is not in the PDB. Did you mean 7Z3Q?
ANSWER
This typo has been corrected
- Line 122 (now line 157): What is “Le Interface analysis?
ANSWER
Typo corrected to “Interface analysis…”
- Line 141 (now line 178): change a-carbons to alpha-carbons. Please clarify whether in A five different models are shown.
ANSWER
a-carbons” corrected to “alpha-carbons”. Caption updated to specify that five species are shown in panel A
The figure legend has been updated as follows:
“Figure 3. Structural alignment and comparative visualization of the leptin–CRH2 complex interfaces across species. (A) Structural superposition of the alpha-carbons of leptin–CRH2 complexes from five species: human (green), mouse (gray), rat (orange), pig (red), and macaque (purple), aligned using the PCMDM template (blue). (B–F) Interface views of the leptin–CRH2 heterodimers for human, mouse, rat, pig, and macaque, respectively. The interface contact surface is shown in cyan. Participating residues from leptin are colored cyan, and those from CRH2 are in yellow and shown as sticks. Residues that are not involved in the human interface but contribute to the interaction in other species are indicated by black arrows. Substituted amino acids that still contribute to the interface are indicated by red arrows. Residues highlighted by black circles participate in the human interface but not in the other species. Residue numbering corresponds to the specific protein sequence of each species.”
- Line 166 (now line 204): What do you mean with “their exploration”
ANSWER
Modified as “Their structural and thermodynamic evaluation”.
- Line 168 (now line 215): What is MDS?
ANSWER
Defined at first mention as “Molecular Dynamics Simulations (MDS)”.
- Line 201 (now line 250): “At 36.5 °C“ the temperature seems irrelevant for the computation of the number of interactions.
ANSWER
In this work we intend to analyse the effect of Temperature in DGb. We studied the number of interactions at 36.5 ºC because it is the physiological temperature of humans, and it could be 1 to 3 degrees higher in other mammals studied in this work. It is important to point out the that the default temperature of most of the programs used here is 25 ºC (298 K). Therefore, we consider relevant to show the interactions at this temperature. We have complemented the table of S2, with the interactions calculated at 37.5 ºC.
- Line 202 (now line 251): What do you mean with “These last explorations”?
ANSWER
The text has been modified, now it reads “These analyses”
- Line 210 (now line 258): 0.15M of what? Is it ionic strength
ANSWER
The manuscript has been updated to clarify that the value refers to ionic strength. The revised sentence now reads:
"At 0.15 M ionic strength (I)”
- Line 216-217 (263-264), Table 2: Please define Kd and Ka; Footnote: Please define I on first use.
ANSWER
The corresponding changes have been made
- Line 239 (now line 292) “The Ka “should be “The predicted Ka”
ANSWER
The correction has been made
- Figure 4: I don't see the point in using the abbreviations h, mac, pig, mou and rat, followed by WT, to refer to the different species. Why not be consistent and use the same nomenclature as in Table 3?
The nomenclature in Figure 4 is now preserved as in Table 3.
- Figure 5: See comments for figure 4 above.
The nomenclature in Figure 5 is now preserved as in Table 3
- Line 378 (now line 433): 3.1. “Thermodynamics Analysis”: The title of this sub-section is clearly wrong.
ANSWER
The appropriate correction has been made, Now it reads “
“3.1. Selection and Preparation of Homologous Sequences”
- Lines 382-383 (now lines 438-440): The UniProt leptin sequences for the five species have 167 aa. However, in Figure 1, leptin sequences have 146 aa. Please describe any existing post-translational modifications. The same could be said for the leptin receptor
ANSWER
We added a note explaining that the sequences in Figure 1 represent mature proteins post signal-peptide removal and may differ from full-length entries. Reference added to UniProt annotations
- Line 395 (now line 452): What are these Two key residues?
ANSWER
The residues N516Q and C604S, have been specified
- Section 3.3 (now section 3.4): Is there any difference between ΔG (line 420, now line 484) and ΔGb (lines 424 and 430, now lines 489 and 493)?
ANSWER
The terminology has been and ensured consistent use of ΔGb for predicted binding free energy.
References issues
- Line 40 (39) : Please check reference 4 (Hübner, 2012) because it does not seem to relate to expression of leptin and Ob-R…
ANSWER
The reference has been replaced. Now the references are [4-6]
- Line 89 (102): Please check reference 28, because it is not related to residue conservation of leptin or Ob-R.
ANSWER
The reference has been replaced. Now the reference is [32]
- Line 103 (127): Please check reference 29, because it seems to only describe a tool to calculate RMSD values.
ANSWER
The reference has been replaced. Now the reference is [35]
- Line 117 (149): I don’t see the relation of references 28 and 31 to the text.
ANSWER
The correction has been made. Now the reference is [36]
- Line 293 (347): Reference 43 deals with thermal stability and not ionic strength. Please insert a suitable reference.
ANSWER
The correction has been made. Now the reference is [22]
- Line 386 (442): Reference 54 is about the “EMBL-EBI Job Dispatcher” service. Although the authors cite (and quite rightly so) this reference for using the online service, in my opinion they should also cite the original reference for the Clustal Omega program (Sievers. and Higgins, 2021)
ANSWER
The correction has been made. Now the reference is [58]
Comments on the Quality of English Language
- The language needs to be improved. Here is a non-exhaustive list of the issues found:
ANSWER
In addition to addressing the reviewer’s specific suggestions, the revised manuscript was professionally edited for English language and clarity using the MDPI English editing service.
- Lines 14-15: It is not clear to me what the authors mean by “While the first interaction between leptin–Ob-R structural aspects have been studied in humans and mice…”, but perhaps it should be rephrased as “While the structural aspects of the interaction between leptin–Ob-R have been first studied in humans and mice…”
ANSWER
The sentence has been replaced as recommended by the revisor
- Line 18: Change "leptin–CRH2" to "leptin–CRH2 complex"
ANSWER
The correction has been made
- Line 19: Change "new template of full-length" to "full-length template"
ANSWER
We have made the suggested correction
- Line 23: its --> is
ANSWER
The correction has been made
- Line 42 (now line 49): have identified --> were identified
ANSWER
The verb has been changed to “have been identified”
- Line 66 (now line 74): “, characterized” --> “and characterize”
ANSWER
The appropriate change has been made
- Line 68 (now line 76) mimic --> mimicking?
ANSWER
We have made the suggested correction.
- Line 85 (98): Please rephrase: “The alignment CRH2 of five Ob-R homologs“
ANSWER
the phrase has been changed to “The CRH2 alignment of the five Ob-R homologs”
- Line 95 (now line 104): leptinCRH2 --> leptin–CRH2
ANSWER
The correction has been made
- Line 125 (now line 162): Serin --> serine.
ANSWER
The change has been made
- Line 172 (now line 211): Perhaps “differences” is a better word than “changes”?
ANSWER
The correction has been made
- Line 186 (now line 235): Please change “, as template in combination of literature” to “as template, in combination with literature”
ANSWER
The change has been made
- Line 200 (now line 249): "In table 1" --> "Table 1"
ANSWER
The change has been made. Now is the Table S2
- Line 216 (now line 262-263): The word “homologous” is misplaced.
ANSWER
The sentence has been change, now it reads
“Table 2. Predicted binding free energy and thermodynamic properties of leptin–CRH2 complexes from five mammalian species.”
- Lines 253-254 (306-309): Please rephrase
ANSWER.
We rephrase the sentences to “The vertical lines in A indicate the pI values of leptin (wide colored lines) and the CRH2 of Ob-R (black crosses), and the vertical lines in B indicate the physiological ionic strength.”
Reviewer 2 Report
Comments and Suggestions for Authors
The hormone leptin is an important protein that plays a role in many physiological processes, and understanding how it is recognized by its receptors is thus a vitally important research area. The leptin receptor Ob-R is found throughout mammalian species, and its uses its CRH2 domain to engage the hormone itself. While structures of the leptin-CRH2 complex have been solved, unresolved regions of the proteins or substitutions may prevent complete understanding of the interactions that drive complex formation.
In this study by Ortega-López et al, homology modeling of the complete leptin Ob-R complex using the precise sequences from five mammalian species was performed, followed by detailed computational analyses to determine the impact of several environmental parameter changes (pH, temperature, ionic strength) on the interaction strength and energetics of this complex. The authors have done very thorough work detailing the precise interactions they observe in the models of each complex, and relate these to the computed interaction strengths and their response to the physicochemical modulations tested. These observations are likely to be very relevant to various physiological stresses a mammalian host could experience, which the authors do a nice job of stressing at the study’s conclusion.
Overall, this reviewer found the ideas and motivation of this work to be sound and interesting. However, there are some key critiques I believe the authors could address to strengthen the manuscript. I think the paper belabors the point that the existing crystallographic models are incomplete, which could be shortened to a single mention. The motivation for carrying out homology modeling being a way to observe differences between the species chosen is already strong enough. Some methodological details should also be more explicitly stated so the reader can understand how the studies were carried out, including the docking of the homology models and the molecular dynamics simulations performed.
Specific critiques:
- The first section of Results could use some additional details as the first sentence (instead of “The text continues here”). Please list the five species you are aligning, and can briefly say “using Clustal Omega” so the reader understands the process.
- Figure 1 appears to be blurry making it hard to read the sequence. Also, the green highlighting is a bit too opaque and obscures the sequence underneath. It is unclear why some amino acids are enclosed in red boxes, not mentioned in the caption.
- PCDMD/PCMDM are used first on lines 94,95. Presumably these should be the same and one is a typo? Since this is the first time you have mentioned this, please explain what it is here, rather than just in the caption of Figure 2.
- The crystal structure of the complex is referred to as PDB ID Z7Q3 on lines 95 and 102, but should be 7Z3Q instead. Also, please make sure to cite the publication of those coordinates, likely on line 92.
- I would tone back the critique of the existing model as your motivation for performing homology modeling. Specifically, the missing residues in CRH2 appear to be far from the interface with leptin, as do the two substitutions. The missing residues in the leptin structure are certainly a significant portion, but these loops are expected to be dynamic. I think stating that the structure provides a nice validation target for your homology modeling, which seeks to compare several species is motivation enough.
- Residues labels in Figure 3 are difficult to read. Could they be made bigger and/or made white with black outline stroke?
- The acronym MDS is first used on line 168 without being defined, please define it. I believe more detail in how these molecular dynamics simulations were performed is imperative for the reader to understand how the models were minimized.
- Table 1 seems quite unwieldy for the main text of a paper. Could this (and possibly Table 3) be moved to supplementary information? The Figures presented already highlight the most important interactions visually in the main text.
- Page 8, line 225. Please include comparison of the values of reported Kd for human complex.
- The pH and ion concentration ranges in Figure 4 seem broader than those in Table 3, i.e. pH 3-10 vs pH 3-8 and 0.05-0.4 M vs 0.05-0.3 M, please be consistent.
- Page 11, line 267. The side chain of Arg has pKa of ~12.5, which seems very far from physiological pH, not sure this should be stated here.
- Page 11, line 284. Was range of ions tested up to 0.45 M? Assuming 4.5 M is a typo.
- Page 13, line 378. Same subheading title is used as previous Results section. Please update to a relevant heading for the methods described (e.g., Sequence Analyses).
- The details of MDS should be given. Which molecular dynamics software was used? With which force field? Was it merely energy minimized, or also equilibrated for some period of time?
- Methods are also unclear as to how the homology models were docked to form the complex. Currently it reads as SWISS-MODEL being used to create homology models of each individual protein with crystal structures as the template. However, it is not specified how the resulting models were then docked to form the complex? Please describe the docking approach.
Overall the English in this manuscript is good. There are a few places where the sentences appear to be incomplete or there are extra words but those can be addressed by editor. A few fixable specific issues:
- In the abstract, the sentence from lines 17-22 goes on quite long; I would recommend breaking this up into multiple sentences for easier readability.
- Page 2, line 74 starts “The text continues here.” Please remove this.
- Page 13, line 353 “Thermodinamics” should be “Thermodynamics”.
- Can remove the “Informed Consent Statement” since it is not applicable to this study.
Author Response
On behalf of the authors, I would like to thank you for the constructive feedback and thoughtful evaluation of our manuscript. We appreciate the time and expertise invested in the review process because all your suggestions help us importantly to improve the clarity of the manuscript. Below, we provide a point-by-point response to each of the comments and suggestions, along with a description of the changes made. All changes in the manuscript are written in red.
Thank you in advance for your kind attention to my request.
The hormone leptin is an important protein that plays a role in many physiological processes, and understanding how it is recognized by its receptors is thus a vitally important research area. The leptin receptor Ob-R is found throughout mammalian species, and its uses its CRH2 domain to engage the hormone itself. While structures of the leptin-CRH2 complex have been solved, unresolved regions of the proteins or substitutions may prevent complete understanding of the interactions that drive complex formation.
In this study by Ortega-López et al, homology modeling of the complete leptin Ob-R complex using the precise sequences from five mammalian species was performed, followed by detailed computational analyses to determine the impact of several environmental parameter changes (pH, temperature, ionic strength) on the interaction strength and energetics of this complex. The authors have done very thorough work detailing the precise interactions they observe in the models of each complex, and relate these to the computed interaction strengths and their response to the physicochemical modulations tested. These observations are likely to be very relevant to various physiological stresses a mammalian host could experience, which the authors do a nice job of stressing at the study’s conclusion.
Overall, this reviewer found the ideas and motivation of this work to be sound and interesting. However, there are some key critiques I believe the authors could address to strengthen the manuscript. I think the paper belabors the point that the existing crystallographic models are incomplete, which could be shortened to a single mention. The motivation for carrying out homology modeling being a way to observe differences between the species chosen is already strong enough. Some methodological details should also be more explicitly stated so the reader can understand how the studies were carried out, including the docking of the homology models and the molecular dynamics simulations performed.
ANSWER
We thank the reviewer for their positive evaluation and thoughtful feedback on the importance and relevance of our study. We appreciate the constructive critiques and have addressed all the suggestions provided. The changes made in response to the reviewer’s comments have significantly improved the clarity, quality, and overall readability of the manuscript.
Specific critiques:
- The first section of Results could use some additional details as the first sentence (instead of “The text continues here”). Please list the five species you are aligning, and can briefly say “using Clustal Omega” so the reader understands the process.
ANSWER
The text has been modified following the reviewer suggestion. Now the first sentence of the first section of Results reads.
To investigate evolutionary conservation and suitability for structural modelling, we first compiled revised leptin and CRH2 sequences from five species (human, mouse, rat, pig, and macaque), from UniProt and aligned them using Clustal Omega. We then constructed homology models for the leptin–CRH2 complexes from five species using validated templates.
- Figure 1 appears to be blurry making it hard to read the sequence. Also, the green highlighting is a bit too opaque and obscures the sequence underneath. It is unclear why some amino acids are enclosed in red boxes, not mentioned in the caption.
ANSWER
We have improved the quality of Figure 1 for clarity. The green highlighting was adjusted to increase sequence readability. Red boxes were removed to avoid confusion, and all relevant annotations are now explained in the caption.
- PCDMD/PCMDM are used first on lines 94,95 (112 and 116). Presumably these should be the same and one is a typo? Since this is the first time you have mentioned this, please explain what it is here, rather than just in the caption of Figure 2.
ANSWER
We corrected the inconsistency and now consistently refer to the model as PCMDM. An explanatory sentence was added at first mention, clarifying that PCMDM refers to the average structure of the main cluster obtained from MD simulations of the human leptin–CRH2 complex.
- The crystal structure of the complex is referred to as PDB ID Z7Q3 on lines 95 (105) and 102 (116), but should be 7Z3Q instead. Also, please make sure to cite the publication of those coordinates, likely on line 92.
ANSWER
We corrected the PDB ID to 7Z3Q throughout the manuscript and included the correct citation at its first mention.
- I would tone back the critique of the existing model as your motivation for performing homology modeling. Specifically, the missing residues in CRH2 appear to be far from the interface with leptin, as do the two substitutions. The missing residues in the leptin structure are certainly a significant portion, but these loops are expected to be dynamic. I think stating that the structure provides a nice validation target for your homology modeling, which seeks to compare several species is motivation enough.
ANSWER
We appreciate this suggestion and have revised the text to attend the critique. The updated version now highlights the utility of the 7Z3Q structure as a reference for validation and emphasizes the primary motivation of our study: comparative modeling across species and under varying environmental conditions.
- Residues labels in Figure 3 are difficult to read. Could they be made bigger and/or made white with black outline stroke?
ANSWER
We have updated Figure 3 with larger, high-contrast labels for improved readability.
- The acronym MDS is first used on line 168 without being defined, please define it. I believe more detail in how these molecular dynamics simulations were performed is imperative for the reader to understand how the models were minimized.
ANSWER
We now define MDS as “molecular dynamics simulations” at its first occurrence in both the Results and Methods sections.
- Table 1 seems quite unwieldy for the main text of a paper. Could this (and possibly Table 3) be moved to supplementary information? The Figures presented already highlight the most important interactions visually in the main text.
ANSWER
As suggested, Table 1 has been moved to the Supplementary Information. But table 3, has not been moved, since we consider the information shown in it is relevant to be show in the manuscript.
- Page 8, line 225 (273). Please include comparison of the values of reported Kd for human complex.
ANSWER
We added a sentence reporting that the predicted Kd for the human complex
- The pH and ion concentration ranges in Figure 4 seem broader than those in Table 3, i.e. pH 3-10 vs pH 3-8 and 0.05-0.4 M vs 0.05-0.3 M, please be consistent.
ANSWER
The pH and ion concentration ranges of Table 3 (Now table 2) has been completed to pH 10 and 0.4 M and is consitent with Figure 4.
- Page 11, line 267 (321). The side chain of Arg has pKa of ~12.5, which seems very far from physiological pH, not sure this should be stated here.
ANSWER
We appreciate the reviewer’s observation. While we agree that the canonical pKa of the arginine side chain is approximately 12.5 in aqueous solution, it is well established that the pKa of ionizable residues can shift significantly depending on the local microenvironment within a folded protein. In some cases, such as within buried or highly interactive regions of the protein–protein interface, pKa values may differ from standard values due to electrostatic interactions, hydrogen bonding, and desolvation effects. Therefore, our statement aimed to highlight the potential role of arginine under specific structural contexts. We have retained the sentence to preserve this point, but would be glad to further clarify or rephrase it if the reviewer or editor might consider it necessary.
- Page 11, line 284 (338). Was range of ions tested up to 0.45 M? Assuming 4.5 M is a typo.
ANSWER
We confirm that this was a typographical error. The correct value is 0.40 M, and it has been corrected in the manuscript.
- Page 13, line 378 (433). Same subheading title is used as previous Results section. Please update to a relevant heading for the methods described (e.g., Sequence Analyses).
ANSWER
We appreciate this observation. The subtitle of this section is now “Selection and Preparation of Homologous Sequences.”
- The details of MDS should be given. Which molecular dynamics software was used? With which force field? Was it merely energy minimized, or also equilibrated for some period of time?
ANSWER
We appreciate the reviewer’s attention to methodological clarity. While we did not perform molecular dynamics simulations ourselves in the present study, the full-length model used as the structural template—referred to as PCMDM—was derived from a previously reported simulation. As now stated more clearly in the revised manuscript, the PCMDM model corresponds to the centroid of the most representative cluster obtained from MD simulations of the human leptin–CRH2 complex (reference included in the updated version) using the OPLS-AA force field of GROMACS. This model was used as a high-quality structural reference for comparative modeling across species.
- Methods are also unclear as to how the homology models were docked to form the complex. Currently it reads as SWISS-MODEL being used to create homology models of each individual protein with crystal structures as the template. However, it is not specified how the resulting models were then docked to form the complex? Please describe the docking approach.
ANSWER
The homology models constructed in this study were generated using SWISS-MODEL. The coordinates of the PCMDM template were obtained as described by Lopez-Hidalgo et al, 2022 [34]. Briefly, the 3D models of leptin and CRH2 were constructed using the I-Tasser server, employing the coordinates of crystallographic structures 1AX8 and 3V6O, respectively. The leptin-CRH2 complex model was obtained by docking on the GRAMM-X server. Subsequently, the complex was subjected to molecular dynamics simulations (MDS) with the GROMACS program for 100 ns, 310 K, and with the OPLS-AA force field. The coordinates used in this work correspond to the average of the most populated group of conformations from the MDS. The manuscript has been modified accordingly to clarify how the homology models were built.
Comments on the Quality of English Language
Overall the English in this manuscript is good. There are a few places where the sentences appear to be incomplete or there are extra words but those can be addressed by editor. A few fixable specific issues:
In addition to addressing the reviewer’s specific suggestions, the revised manuscript was professionally edited for English language and clarity using the MDPI English editing service.
- In the abstract, the sentence from lines 17-22 goes on quite long; I would recommend breaking this up into multiple sentences for easier readability.
ANSWER
The correction has been made. Now it reads “We performed a bioinformatics-driven structural, stability, and thermodynamic characterization of the leptin–CRH2 complex. This included structural homology modeling using a full-length template, interface mapping, and binding energy estimation. Additionally, we analyzed the effect of pH, ionic strength, and temperature on complex formation to mimic physiological and pathological tissue conditions.”
- Page 2, line 74 (82) starts “The text continues here.” Please remove this.
ANSWER
This phrase has been removed
- Page 13, line 353 (408) “Thermodinamics” should be “Thermodynamics”.
ANSWER
The correction has been made
- Can remove the “Informed Consent Statement” since it is not applicable to this study.
ANSWER
The Informed Consent Statement, has been eliminated
Round 2
Reviewer 1 Report
Comments and Suggestions for Authors
The authors addressed all of the reviewer's comments, and the text is now clearer. It also became clear what constitutes prior work and what was actually done in this study. In my opinion, using a single model obtained by protein-protein docking and subsequent molecular dynamics simulations as a template for obtaining models for the five mammalian species is a somewhat fragile approach. An approach that used multiple templates for the starting structure would have been preferable: well-resolved models of leptin and the receptor separately, and the incomplete model of the complex, which would provide the "correct" orientation of leptin relative to its receptor. However, the authors' approach also makes some sense, and therefore, in this second review of the paper, I have only a few minor issues to point out..
Minor issues
The authors state that Gln516 and Cys604 are located near the leptin-binding interface, implying that this is the leptin-receptor interface. In reality, these two residues are not close to this interface, as can be seen in Figure 1. Cys604, in particular, appears to be involved in interactions with another domain of the receptor (see Tsirigotaki et al, 2023) and far from leptin.
Please be consistent with the designation of the helical elements of leptin. In line 41, helices are identified as A-D; in Figure 1.A, as H1-H4; in line 157, as a1 and a3; in line 185, as 1 and 3. Please choose one notation and follow it throughout the text.
In lines 112-113, the authors state that the PCMDM model “corresponds to the average structure of the most representative cluster“ but in lines 467-468, it corresponds “to the centroid structure of the largest conformational cluster”, which are different things. Please correct this.
In line 139, in reality, the PCMDM is shown in blue and yellow
In lines 226 and 229, it appears that some closing parentheses are missing after “macaque”
The vertical line in Figure 5 has disappeared in this revision.
The authors now provide the coordinates of the starting model used in this work. However, these are presented in the form of a long table, which is rather cumbersome. Please provide a standard PDB file as supplementary material instead.
Author Response
MDPI Manuscript ID: ijms-3715096
Title: Bioinformatic analysis of the leptin–Ob-R interface: structural modeling, thermodynamic profiling, and stability in diverse microenvironments
Reviewer 1
On behalf of the authors, I would like to thank you again for the detailed revision and constructive feedback on our manuscript. All your comments have been very valuable for improving our manuscript. Below, we provide a point-by-point response to each of the comments and suggestions raised by the reviewers, along with a description of the changes made. All changes in the manuscript are written in red.
Thank you in advance for your kind attention.
Yours sincerely,
Reviewer 2 Report
Comments and Suggestions for Authors
The revisions the authors have made have greatly improved the manuscript, and I appreciate the thorough responses to my previous comments. I believe it is now acceptable in its present form. I do have a few suggested minor changes if the authors see fit to incorporate for the final version.
1. The residue labels on Fig 3, panels C-F do not seem to have been enlarged/improved as stated. The ease of interpretation of this figure would be greatly enhanced by larger, higher contrast residue number labels if possible.
2. While a trivial conversion, it is a little disorienting to refer to temperatures in both units of °C and K. Maybe as a compromise, put the corresponding temperature range in parentheses? Specifically, page 12 line 356.
Author Response
MDPI Manuscript ID: ijms-3715096
Title: Bioinformatic analysis of the leptin–Ob-R interface: structural modeling, thermodynamic profiling, and stability in diverse microenvironments
Reviewer 2
On behalf of the authors, I would like to thank you again for the detailed revision and constructive feedback on our manuscript. All your comments have been very valuable for improving our manuscript. Below, we provide a point-by-point response to each of the comments and suggestions raised by the reviewers, along with a description of the changes made. All changes in the manuscript are written in red.
Thank you in advance for your kind attention.
Yours sincerely,